# Flow signal change in polyps after anti-vascular endothelial growth factor therapy

Chia-Jui Chang[1]☯, Yi-Ming Huang[1]☯, Ming-Hung Hsieh[2]☯, An-Fei Li[1,3,4]☯, Shih-Jen Chen (ID)[1,3]☯ *

1 Department of Ophthalmology, Taipei Veterans General Hospital, Taipei, Taiwan, 2 Hoping Branch, Taipei City Hospital, Taipei, Taiwan, 3 National Yang-Ming University School of Medicine, Taipei, Taiwan, 4 Department of Ophthalmology, Cheng Hsin General Hospital, Taipei, Taiwan

☯ These authors contributed equally to this work.
* sjchen96@gmail.com

**Data Availability Statement:** All relevant data are within the manuscript.

**Funding:** The authors received no specific funding for this work.

## Abstract

Optical coherence tomography angiography (OCTA) is a novel, non-invasive imaging tool used to detect vascular flow. The absence of a flow signal in OCTA in polyps revealed by indocyanine green angiography (ICGA) in patients with polypoidal choroidal vasculopathy (PCV) may indicate slow or compromised filling of blood flow from choroidal vessels. Naïve patients with PCV treated with intravitreal injections of aflibercept (IVI-A) were enrolled in this study to validate the hypothesis that baseline flow may affect the outcome of polyp regression in ICGA. The flow signal of polyps in OCTA was detected by manual segmentation in the corresponding location by ICGA. Polyps were defined as high-flow if both OCTA and ICGA showed positive findings, and low-flow if OCTA showed a negative flow signal in 3 consecutive horizontal scans at the polyp area shown in ICGA. A total of 24 polyps were identified in 13 PCV patients at baseline. Of these 24 polyps, 22 (91.7%) were high-flow and 2 (8.3%) were low-flow. After 3 monthly IVI-A, all low-flow polyps had complete regression in ICGA. Among 17 (77%) high-flow polyps at baseline that had regression after treatment, 10 (58.8%) became low-flow, while 5 (22.7%) persistent polyps remained high-flow. Flow signal of polyps as detected by OCTA could be a predictive factor for treatment response in patients with PCV. Monitoring changes in flow signal after treatment is clinically relevant.

## Introduction

Optical coherence tomography angiography (OCTA) is a new and non-invasive imaging tool used to detect vascular flow by computing signal decorrelation from moving blood in vessels between consecutive cross-sectional scans. Compared to the gold standard of indocyanine green angiography (ICGA) [1], OCTA has a greater branching vascular network (BVN) detection rate of 80~100% but lower polyp detection rate of 45~85% [2–10]. Applying OCTA in addition to fundus color imaging, fluorescein angiography and structural OCT has been shown to increase the sensitivity but not the specificity of diagnosing polypoidal choroidal vasculopathy (PCV) [11]. This may be due to the over diagnosis of occult choroidal neovascularization (CNV) as PCV caused by the low polyp detection rate of OCTA.

**Competing interests:** No authors have competing interests.

Potential reasons for the low detection rate of polyps in OCTA include the extra-macular location of polyps outside the central 6x6 mm area, segmentation errors, especially when there is highly elevated retinal pigment epithelium (RPE) detachment, and low vascular flow within the polyps [12]. Several studies have reported that the polyp detection rate of OCTA can be improved by adjusting the segmentation manually [5–7, 9, 10]. OCTA can detect flow with a velocity as low as 0.2~0.3 mm/s using a 70k-Hz A-scan within seconds [13–15]. In contrast, ICGA takes at least 5 minutes to identify polyps from the first 30 seconds of video recorded by a confocal scanning laser ophthalmoscope in the filling phase of the polyps, and 5 minutes to picture hyper-fluorescent polyps in the early phase [16, 17]. Rebhun et al. found that the flow velocity not only varied from patient-to-patient but also polyp-to-polyp [10]. Huang et al. used OCTA to characterize the morphology of BVN, and found that different BVN patterns had different recurrence and response rates to the treatment [8]. Since different polyps may have different flow velocities and filling times as reflected by the detection ability of OCTA in comparison to ICGA, we hypothesized that polyps with different flow rates may have different responses and outcomes to anti-vascular endothelial growth factor (anti-VEGF) treatment. Based on this hypothesis, this study first aimed to categorize polyps into high flow, which could be detected by both ICGA and OCTA, and low flow, which could only be detected by ICGA. The second aim was to evaluate the outcomes of the polyps after anti-VEGF treatment based on their flow type.

## Materials and methods

This study was reviewed and approved by the Institutional Review Board of Taipei Veterans General Hospital (Taipei, Taiwan, approval ID: 2018-09-006BC), and it was performed according to the relevant guidelines and regulations. Informed consent was obtained from all participants in this study. Patients with naïve PCV diagnosed by ICGA were consecutively enrolled from September 2018 to August 2019 at a tertiary medical center in Taiwan. Patients who had ever received treatment including intravitreal anti-VEGF injections or photodynamic therapy (PDT) before enrollment were excluded. In addition, patients with poor image quality due to fixation loss, medium opacity, massive hemorrhagic PED or submacular hemorrhage obscuring the polyp were also excluded. ICGA (Heidelberg Engineering Inc., Heidelberg, Germany) was performed at baseline and after 3 monthly intravitreal injections of aflibercept (IVI-A). Early and mid-phase ICGA were read by two retinal specialists separately (AFL and SJC), who were blinded to each other's diagnosis and the clinical information of the patients. Lesions were enrolled only when they were considered to be polyps by both retinal specialists. The response after three IVI-A were determined only when there was consensus between the two retinal specialists. The response in ICGA was classified as complete regression when the nodular hyperfluorescence of the polyp disappeared, partial regression when the polyp shrunk in size, and persistent when the polyp remained at the pre-treatment size. If there was any disagreement on the diagnosis or response of the polyps in ICGA, the 2 retinal specialists discussed and if a consensus could not be reached, the polyp was excluded for the OCTA study. Polyps beyond the central 6x6 mm image of ICGA were also excluded.

The patients underwent spectral domain-OCT (SD-OCT) (Optovue, Fremont, CA) to evaluate double layer signs of BVN, bumps or notches of pigment epithelium detachment (PED) for the polyps, and subretinal fluid (SRF) at baseline and after treatment. OCTA (Avanti; Optovue, Fremont, CA) was performed after the structural OCT to evaluate the polyps and BVN. Polyps within 3x3 or 6x6 mm of the central macula were counted and compared to the flow signal beneath the RPE by OCTA. The flow signal was detected by manual segmentation from the superficial retinal layer to choriocapillaries in the corresponding location appearing

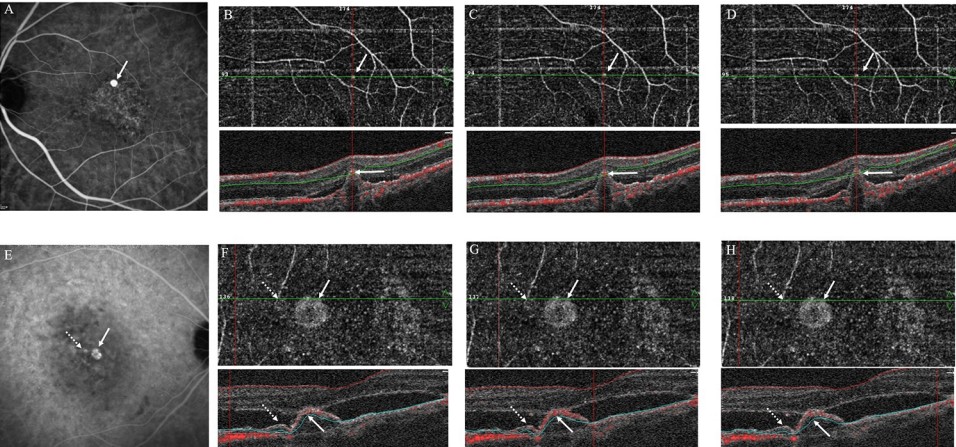

**Fig 1. ICGA and OCTA of high-flow polyp and low-flow polyp.** (A) ICGA revealed one polyp (arrow). (B~D) The polyp (arrow) was identified by en face flow OCTA, and flow signal (arrow) was identified by en face flow OCTA, and flow signal was detectable in all 3 consecutive horizontal scans at the center of the polyp. The polyp was then defined as "high flow." (E) Two polyps identified on ICGA (arrow and dotted arrow). (F~H) En face OCTA and 3 consecutive horizontal scan of the 2 types of polyps. The flow signal of "high flow" polyp (arrow) was detectable in en face and all 3 consecutive horizontal scans, while the flow signal of the "low flow" polyp (dotted arrow) could not be detected in en face and all 3 horizontal scans.

in ICGA. Projection artifacts from the retinal vessels were carefully examined and excluded. The OCTA manual segmentation of each polyp was then done by a third specialist (YMH). This third specialist interpret the signal of OCTA at the polyps where the 2 specialists agreed on. The En-face OCTA and horizontal OCTA scans were both used for flow signal detection. The signal of En-face OCTA was studied for flow signal at areas of polyps and BVN shown in ICGA. To confirm the flow signal, especially in equivocal signal after anti-VEGF, horizontal OCTA scan were used line by line at polyps for confirmation. Three sequential horizontal scans at the center of the corresponding area of the polyps in ICGA were performed for all polyps. These 3 horizontal scans covered a vertical size of 30 μm in 3x3 mm and 45 μm in 6x6 mm OCTA. The flows were categorized into high flow (presence of flow signal in at least 3 consecutive horizontal scans), low flow (absence of flow signal in all 3 consecutive horizontal scans) (Fig 1), and equivocal flow (presence of flow signal in 1 or 2 of 3 consecutive horizontal scans). The area of the post-treatment OCTA was evaluated according to the same area and size (either 3x3 mm or 6x6 mm) at baseline, e.g., 3x3 mm at both baseline and after treatment. Polyps accompanied with or without PED were also assessed.

Snellen visual acuity (VA) was transformed to logMAR for analysis. Statistical analyses were performed using SPSS software version 12.0 (SPSS Inc., Chicago, IL, USA), using the Student's t-test. A p value $< 0.05$ was considered to be statistically significant.

## Results

During this study time period, a total of 62 patients with PCV were initially enrolled. Twenty-four of them were excluded because of the history of previous treatment with intravitreal anti-VEGF injections and PDT. Twenty-five patients were further excluded due to poor image quality by massive hemorrhage, hemorrhagic PED, or poor fixation. In the end, 13 patients (8 males and 5 females) were enrolled with an average age of 66.69±9.22 years and average visual acuity of 0.37±0.16 logMAR (Snellen equivalent of 0.45±0.176). A total of 24 polyps were identified by ICGA before IVI-A. The number of polyps ranged from 1 to 4 for each patient, with

**Table 1. Demographic data and clinical profile of the 24 polyps in 13 patients before and after treatment.**

| Case | Sex | Age | VA | SRF | Polyp | ICGA | Flow in OCTA | PED | BVN (size, DA) | VA | SRF | Outcome in ICGA | Flow in OCTA | PED | BVN (size, DA) |
|------|-----|-----|-----|-----|-------|------|--------------|-----|----------------|-----|-----|-----------------|--------------|-----|----------------|
| | | | | | | | Before Treatment | | | | | After Treatment | | | |
| 1 | F | 73 | 6/7.5 | + | 1 | + | Positive | + | 1.0 | 6/6 | - | Complete | Equivocal | + | 0.75 |
| | | | | | 2 | + | Positive | + | | | - | Partial | Positive | + | |
| | | | | | 3 | + | Positive | + | | | - | Complete | Positive | + | |
| 2 | M | 78 | 6/30 | + | 4 | + | Positive | + | - | 6/7.5 | - | Complete | Negative | + | - |
| 3 | M | 75 | 6/15 | + | 5 | + | Negative | + | - | 6/15 | + | Complete | Negative | - | - |
| | | | | | 6 | + | Positive | + | | | + | Partial | Negative | + | |
| 4 | M | 69 | 6/15 | + | 7 | + | Positive | + | 2.0 | 6/8.6 | - | Partial | Equivocal | + | 2.0 |
| 5 | M | 64 | 6/8.6 | + | 8 | + | Positive | + | 2.0 | 6/7.5 | - | Partial | Equivocal | + | 2.0 |
| 6 | F | 57 | 6/15 | + | 9 | + | Positive | + | 1.0 | 6/8.6 | - | Complete | Negative | + | 1.0 |
| | | | | | 10 | + | Positive | + | | | - | Persistent | Positive | + | |
| | | | | | 11 | + | Positive | + | | | - | Persistent | Positive | + | |
| | | | | | 12 | + | Positive | + | | | - | Complete | Negative | - | |
| 7 | M | 57 | 6/15 | + | 13 | + | Positive | + | 1.0 | 6/6.7 | - | Persistent | Positive | + | 1.0 |
| 8 | M | 62 | 6/20 | + | 14 | + | Positive | + | - | 6/20 | + | Persistent | Positive | + | - |
| | | | | | 15 | + | Positive | + | | | + | Partial | Positive | + | |
| 9 | F | 61 | 6/20 | + | 16 | + | Positive | + | 1.0 | 6/10 | - | Complete | Negative | + | 0.75 |
| | | | | | 17 | + | Negative | - | | | - | Complete | Negative | - | |
| 10 | M | 58 | 6/10 | + | 18 | + | Positive | + | 3.0 | 6/6.7 | - | Persistent | Positive | + | 3.0 |
| 11 | F | 86 | 6/15 | + | 19 | + | Positive | + | - | 6/10 | - | Complete | Negative | + | - |
| 12 | M | 69 | 6/10 | + | 20 | + | Positive | + | 1.0 | 6/7.5 | - | Partial | Positive | + | 1.0 |
| | | | | | 21 | + | Positive | + | | | + | Partial | Positive | + | |
| | | | | | 22 | + | Positive | + | | | - | Partial | Positive | + | |
| 13 | F | 58 | 6/15 | + | 23 | + | Positive | + | 0.75 | 6/8.6 | - | Complete | Negative | + | 0.75 |
| | | | | | 24 | + | Positive | + | | | - | Partial | Positive | + | |

BVN: branching vascular networks; DA: disc area; ICGA: indocyanine green angiography; OCTA: optical coherence tomography angiography; PED: pigment epithelium detachment; SRF: subretinal fluid; VA: visual acuity.

an average of 1.84 polyps per case. All patients had SRF at the fovea as detected by SD-OCT before treatment (Table 1).

At baseline, among the 24 polyps identified by ICGA, 22 (91.7%) which had a detectable flow in at least three horizontal scans were categorized as being high flow by OCTA. In addition, two (8.3%) polyps which had undetectable flow in all horizontal scans were categorized as being low flow. None of the polyps were classified into the equivocal-flow group at baseline (Fig 2).

All polyps were located beneath the RPE. All polyps had accompanying PED, except for 1 low-flow polyp (polyp 17). Nine among 13 patients had BVN before the treatment. The size of BVN ranged from 0.75 to 3.0 disc area(DA) in ICGA (average of 1.42±0.75 DA).

After 3 consecutive intravitreal injections of aflibercept, the average visual acuity improved to 0.17±0.15 logMAR (Snellen equivalent of 0.71±0.20) (p<0.001). In the structural OCT, 19 (79.2%) polyps showed no SRF after treatment on OCT, while 5 (20.8%) had persistent SRF. For the 23 polyps with PED at the baseline, 21 (91.3%) showed persisted but smaller PED and 2 (8.7%) showed complete resolution of the PED after treatment. After treatment, the size of the BVN remained unchanged (1.36±0.79 DA, p = 0.88) in our short term 3 months follow up.

As shown in Fig 2, ICGA revealed that 10 (41.7%) of the 24 polyps had complete regression, 9 (37.5%) had partial regression, and 5 (20.8%) remained persistent after treatment. While

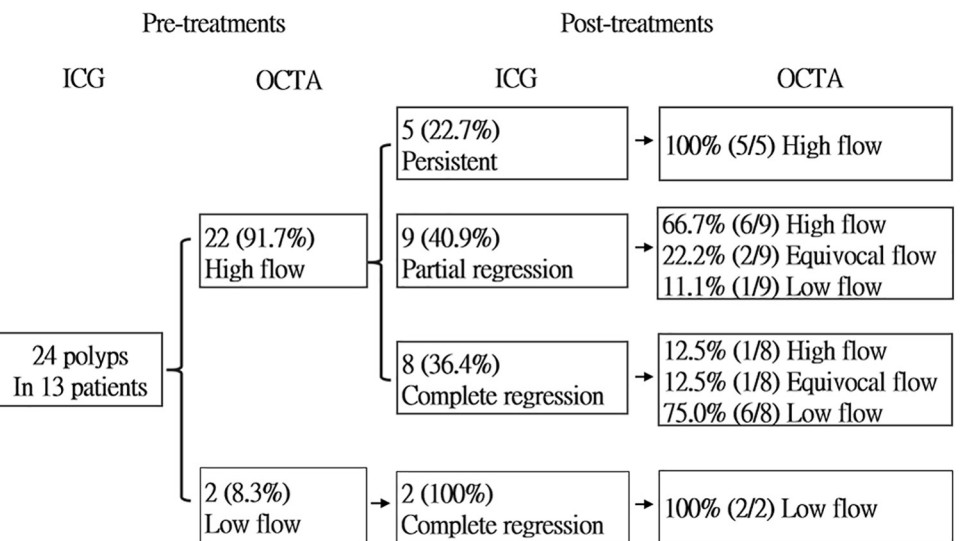

**Fig 2. Outcomes of the 24 polyps detected in ICGA and flow signal changes in OCTA before and after 3 monthly loading of intra-vitreal injections of aflibercept (IVI-A).**

OCTA revealed that 12 (50.0%) polyps showed high flow, 3 (12.5%) showed equivocal flow, and 9 (37.5%) showed no flow signal (Figs 3–5).

Both of the low-flow polyps at baseline had complete regression in ICGA and the flow remained undetectable in OCTA. For the 22 high-flow polyps at baseline, 8 (36.4%) had complete regression, 9 (40.9%) had partial regression, and 5 (22.7%) remained persistent in ICGA. Of these 22 high-flow polyps, 12 (54.5%) remained high-flow, 3 (13.6%) became equivocal-flow, and 7 (31.8%) became low-flow in OCTA after treatment.

Among the 17 polyps that showed complete or partial regression in ICGA, the flow could still be detected in 58.8% (10/17) in OCTA.

For the 12 polyps that remained high flow after treatment, 9 (75.0%) showed resolved SRF and 3 (25.0%) showed persistent SRF. Although PED persisted in all 12 polyps, 10 (83.3%) showed an improvement in the size of the PED.

SRF resolved in all 3 polyps that changed from high flow to equivocal flow after treatment. Although PED persisted in all 3 polyps, all of them became smaller after treatment.

Of the 7 polyps that changed from high to low flow after treatment, SRF resolved in 6 (85.7%) but persisted in 1 (14.3%). PED persisted but improved in all 7 of these polyps.

## Discussion

Our results showed that the presence or absence of a flow signal in the polyps in OCTA may have been associated with different treatment responses after intravitreal anti-VEGF injections. Both of the 2 low-flow polyps in OCTA had complete regression in ICGA after treatment, while 77.3% of the 22 high-flow polyps had complete or partial regression in ICGA. Among these polyps with complete or partial regression, 58.8% had no detectable or equivocal flow signal in OCTA. All of the 5 (22.7%) persistent polyps in ICGA had a detectable flow, although SRF resolved in 4 (80%) of them. Differences in flow rate between polyps may indicate the heterogeneity of polyps in response to anti-VEGF or even to photodynamic therapy. Furthermore, our study also found that the size of the BVN in ICGA remained unchanged after treatment in our short term 3 months follow up. This was in accordance with previous studies that the BVN had poorer response to anti-VEGF than the polyps did [16, 18–22].

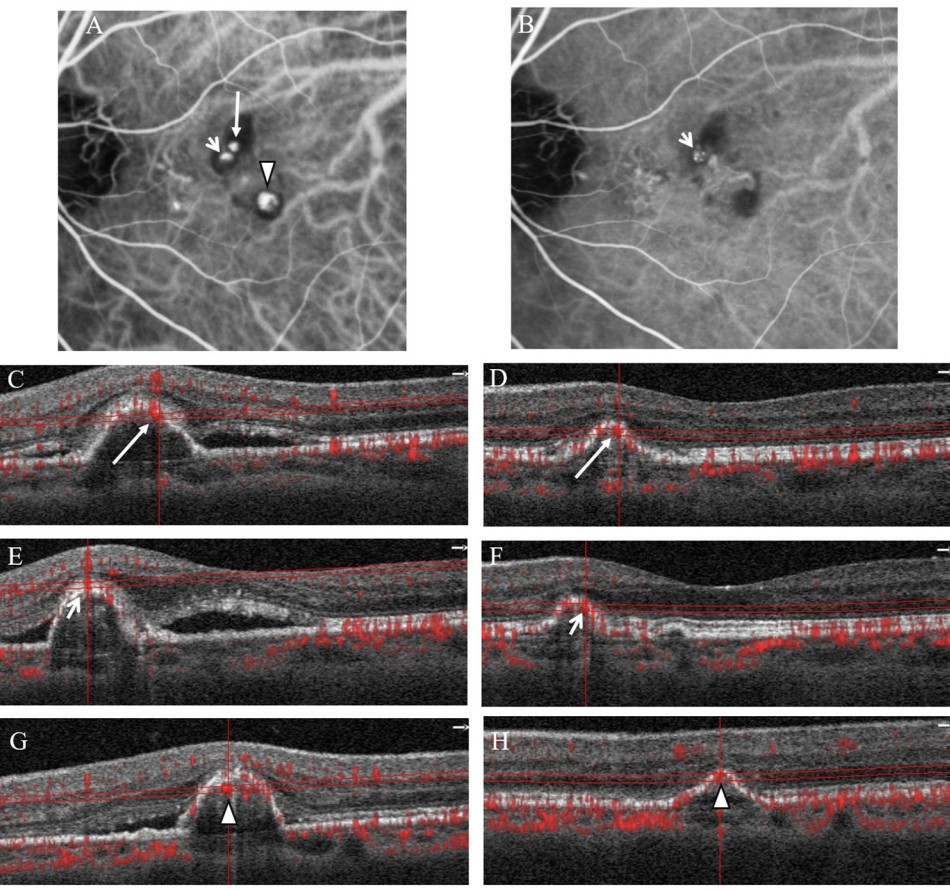

**Fig 3. ICGA and OCTA of case 1.** (A) ICGA before treatment showed 3 polyps, labeled as polyp 1 (long arrow), polyp 2 (short arrow), and polyp 3 (arrowhead). (B) ICGA after treatment showed complete regression of polyp 1 and 3, but partial regression in the size of polyp 2 (short arrow). (C) OCTA of polyp 1 before and (D) after treatment showed the presence of flow signal (long arrow). (E) OCTA of polyp 2 before and (F) after treatment showed the presence of flow signal (short arrow). (G) OCTA of polyp 3 before and (H) after treatment showed the presence of flow signal (arrowhead).

Polyps regression as a treatment goal for PCV management remained controversial since recent clinical trials had shown improved visual acuity with inactive but persisted polyps [18, 22]. A recent study with up to 11 years of follow-up showed that the long-term risk for massive submacular hemorrhage was significantly higher in patients with persistent polyps than those without [23]. The 2 years EVEREST II and the 2 years PLANET studies reported that the percentage of complete polyp regression as assessed by ICGA was around 34.7% after intravitreal injections of ranibizumab and 38.9% after aflibercept monotherapy at 12 months [18, 22]. The number of injections in the monotherapy arm ranged from 3 to 12, indicating that a high variety of polyps responded to anti-VEGF treatment. Even with combination therapy, 30% of the polyps remained present at 12 months [18]. Although several studies reported that single polyps, a smaller polyp area, and non-cluster type polyps had a better response after anti-VEGF monotherapy [19–21], it remained unknown whether the flow rate played a role in the treatment response. Our study showed that flow signal in OCTA, which may be attributed to the blood flow rate from the choroid to the polyps [12], could be a possible reason for the variable response to anti-VEGF treatment.

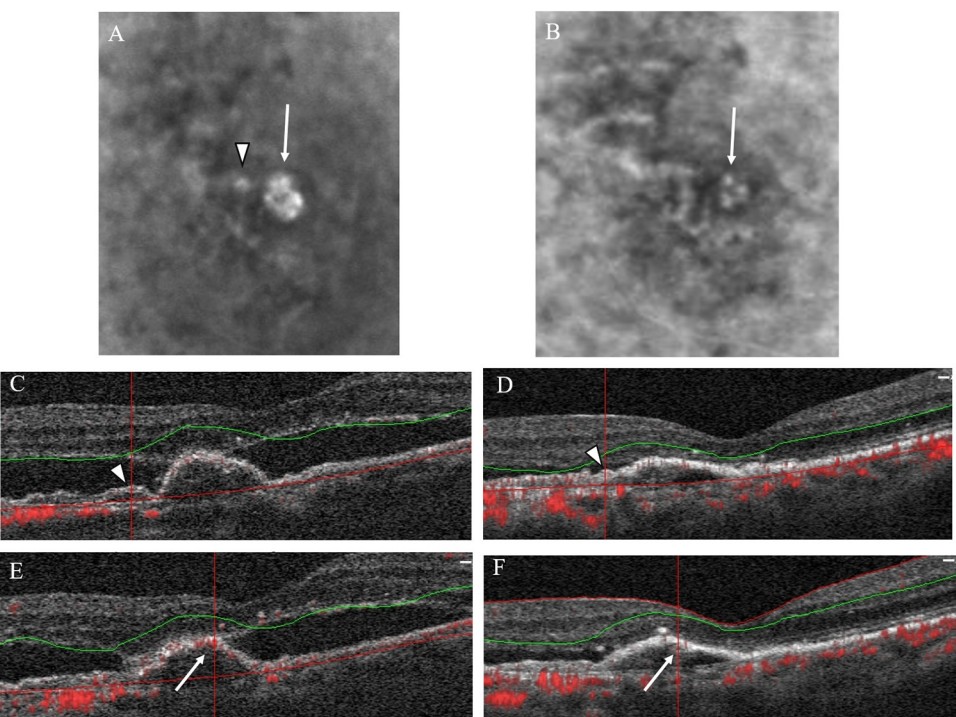

**Fig 4. ICGA and OCTA of case 3.** (A) ICGA before treatment showed 2 polyps, labeled as polyp 5 (arrowhead) and polyp 6 (arrow). (B) ICGA after treatment showed complete regression of polyp 5, but partial regression of polyp 6 (arrow). (C) OCTA of polyp 5 before and (D) after treatment showed the absence of flow signal (arrowhead) in this low-flow polyp. (E) OCTA of polyp 6 before treatment showed the presence of flow signal (arrow) but (F) the absence of flow signal (arrow) after treatment. Note the decrease in subretinal fluid and height of pigment epithelial detachment after treatment.

The presence of flow signal in OCTA could also be associated with clinical signs. Of the polyps with no flow signal or with an equivocal flow signal in OCTA after treatment, 83% (10/12) had complete polyp regression and anatomical improvement with disappearance of the SRF and decreased height of the PED. Moreover, of the polyps that still had high-flow in OCTA, 42% (5/12) persisted without becoming smaller, and 25% (3/12) had persisted SRF at the fovea. In addition to the activity of the polyps as revealed by the presence of SRF after 3 initial loading injections of anti-VEGF, a change in flow signal from high to low in OCTA may have predicted a better response with regards to polyp regression and clinical improvement. Moreover, the responses of the naïve polyps with no flow signal at baseline were even better. Our results suggest that flow signal of individual polyps in OCTA at baseline and follow-up after anti-VEGF treatment could be a prognostic factor for treatment outcomes and may alleviate the need for frequent ICGA studies.

It should be noted that 2 among 8 completely regressed polyps in ICGA after treatment still had flow signal in OCTA. Theoretically, the OCTA scan could detect very slow flow signal with velocity of 0.2 to 3 mm/s [13–15]. Therefore, though the ICGA after the treatment revealed complete regression of the polyp, there might be slow flow remaining in the area that could be defected by the OCTA but not visualized in ICGA.

There are several limitations to this study. First, the numbers of patients and polyps were small, especially for the low flow polyps, and the follow-up time was short. Because of the wide detection of flow signal by OCTA, the low flow polyps may indeed the polyps with very low

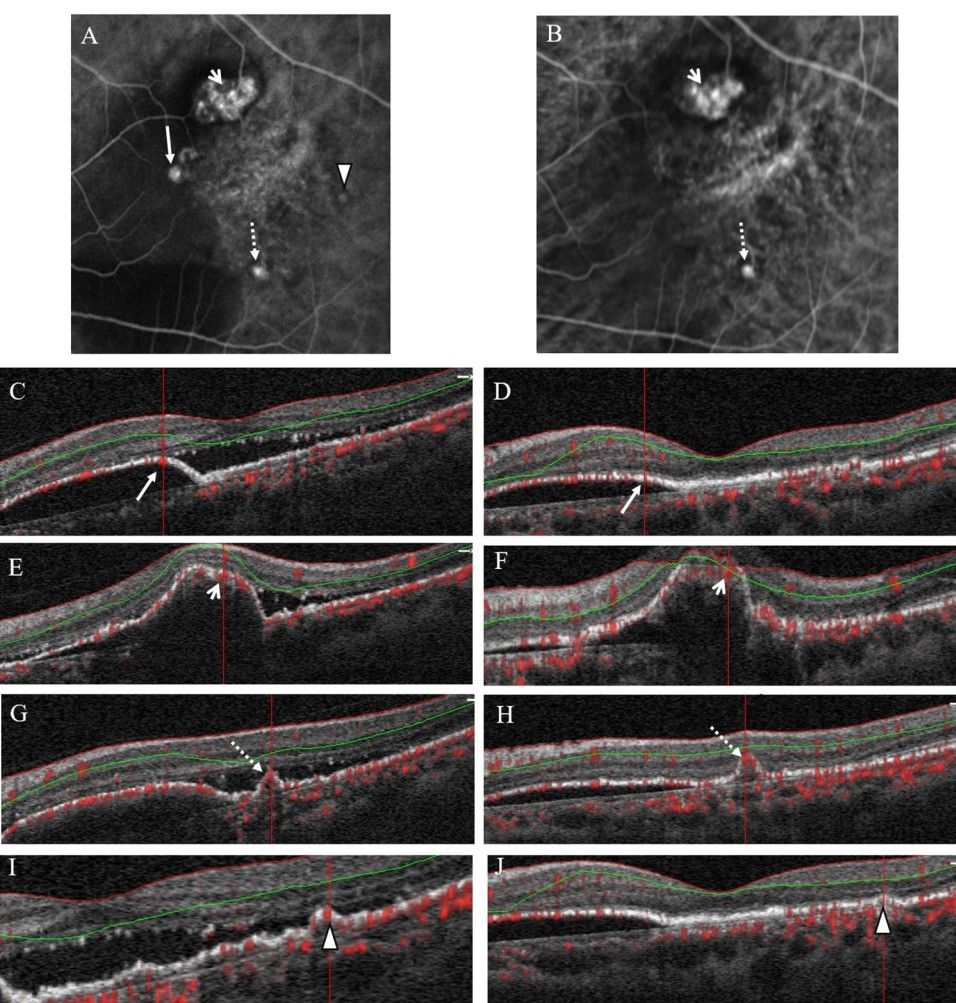

**Fig 5. ICGA and OCTA of case 6.** (A) ICGA before treatment showed 4 polyps, labeled as polyp 9 (long arrow), polyp 10 (short arrow), polyp 11 (dotted arrow) and polyp 12 (arrowhead). (B) ICGA after the treatment showed complete regression of polyp 9 and 12, but the persistence of polyp 10 (short arrow) and polyp 11 (dotted arrow). (C) OCTA of polyp 9 before treatment showed the presence of flow signal (arrow) but (D) the absence of flow signal (arrow) after treatment. Note that the adjacent signal was projection artifacts from the retinal vessels. (E) OCTA of polyp 10 before and (F) after treatment showed the presence of flow signal (short arrow). (G) OCTA of polyp 11 before and (H) after treatment showed the presence of flow signal (dotted arrow). (I) OCTA of polyp 12 before treatment showed the presence of flow signal (arrowhead) but (J) the absence of flow signal (arrowhead) after treatment.

filling flow of the polyps and the number may be quite few in symptomatic patients with PCV. It will be interesting to see if there are more low flow polyps in asymptomatic patients with PCV. Second, inaccuracies in flow signal detection were possible due to the manual measurements. However, by excluding the polyps with an equivocal diagnosis, meaning that suspicious lesions were considered to be polyps by one retinal specialist but not by another retinal specialist who interpreted the same ICGA separately, we believe that the rate of inaccurate detection was decreased. Third, we only recorded the presence or absence of flow signal in the polyps, but we did not quantify the flow intensity in the polyps or BVN in OCTA as this would have involved more sophisticated analysis of a 3-D structure of the lesions. And lastly, this classification of flow and response of polyps might not be applied to all kinds of patients with PCV due

to the exclusion of patients with previous treatment of anti-VEGF or PDT or patients without good image quality of OCTA such as massive hemorrhage or poor fixation.

In conclusion, flow signal in polyps as revealed by OCTA could be associated with the response to anti-VEGF treatment in patients with PCV, and a low flow signal in polyps may indicate a good response. In addition, monitoring changes in the signal after treatment is clinically relevant and could be considered in the management of patients with PCV in the future.

## Author Contributions

**Investigation:** An-Fei Li.

**Methodology:** Shih-Jen Chen.

**Resources:** Yi-Ming Huang, Ming-Hung Hsieh, An-Fei Li.

**Supervision:** Shih-Jen Chen.

**Writing – original draft:** Chia-Jui Chang.

**Writing – review & editing:** Yi-Ming Huang, Shih-Jen Chen.

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
