## [Decision Letter · Decision Letter 0]

11 May 2020

PONE-D-20-09993

Flow Signal Change in Polyps After Anti-Vascular Endothelial Growth Factor Therapy

PLOS ONE

Dear Dr Chen,

Thank you for submitting your manuscript to PLOS ONE. After careful consideration, we feel that it has merit but does not fully meet PLOS ONE’s publication criteria as it currently stands. Therefore, we invite you to submit a revised version of the manuscript that addresses the points raised during the review process.

We would appreciate receiving your revised manuscript by Jun 25 2020 11:59PM. To enhance the reproducibility of your results, we recommend that if applicable you deposit your laboratory protocols in protocols.io, where a protocol can be assigned its own identifier (DOI) such that it can be cited independently in the future. For instructions see: http://journals.plos.org/plosone/s/submission-guidelines#loc-laboratory-protocols

We look forward to receiving your revised manuscript.

Kind regards,

Pukhraj Rishi, MD

Academic Editor

PLOS ONE

Additional Editor Comments (if provided):

This is an interesting paper addressing a relevant clinical issue that needs further investigation and analysis.

The reviewers have raised some very relevant points including changes in the methodology that will help enhance the quality of the manuscript.

2. Thank you for stating the following financial disclosure: "No"

Reviewers' comments:

Reviewer's Responses to Questions

**Comments to the Author**

1. Is the manuscript technically sound, and do the data support the conclusions?

Reviewer #1: Partly

Reviewer #2: Yes

Reviewer #3: Yes

2. Has the statistical analysis been performed appropriately and rigorously? 

Reviewer #1: Yes

Reviewer #2: Yes

Reviewer #3: Yes

3. Have the authors made all data underlying the findings in their manuscript fully available?

Reviewer #1: No

Reviewer #2: Yes

Reviewer #3: Yes

4. Is the manuscript presented in an intelligible fashion and written in standard English?

Reviewer #1: Yes

Reviewer #2: Yes

Reviewer #3: Yes

5. Review Comments to the Author

Reviewer #1: 1. Kindly explain why horizontal OCTA scans have been preferred over En-face OCTA scan.

2. Presence of Branching vascular network (BVN) complex has not been recorded. Any correlation between polyps and BVN complex could also have been identified.

3. Fig 1 (F, G, H) showing En-face OCTA of low flow polyp, could not be considered a proper scan. The manual segmentation is not at the appropriate level to be able to visualise the polyp.

4. Figure 2 is incomplete. No details of high flow polyps given.

5. Figure 3 depicting ICGA after treatment, shows complete remission of polyp 1 and 3. However, flow signals in the polyps still persist on OCTA after treatment. How can it be explained?

6. A sample of 2 low flow polyps is clearly inadequate to conclude the relevance or to predict clinical response to anti-VEGF.

Reviewer #2: Comments:

The authors of this manuscript are interested to study the dynamic change of blood flow within polypoidal lesions and whether it is a predictive factor for PCV response to intravitreal anti-VEGF treatment. Regression and inactivity of polypoidal lesions are considered as important clinical outcomes in multicentre randomized controlled trials of intravitreal anti-VEGF in PCV eyes. Hence, the authors had picked a highly relevant research topic with a sound hypothesis.

OCTA may not be as sensitive as ICGA for the detection of polyopidal lesions, and the authors had already acknowledged a number of intrinsic limitations of current OCTA imaging technology. The examination of 3 sequential OCTA scans in this study had somewhat reduced the error rate in imaging analyses. Nevertheless, segmentation error could be commonly encountered in PCV eyes due to irregularities at the level of retinal pigment epithelium and large, tall pigment epithelial detachments. A large amount of macular haemorrhage and exudation may also block OCT signals. PCV eyes with poor visual acuity may not be able to fixate well during OCTA scan leading to motion artefacts. Were there any eyes excluded from the study and what were the reason for their exclusion? I believe providing details on the exclusion criteria and the number of patients that were excluded from the consecutive recruitment process is very important as it conveys to the readers how practical it would be to examine PCV eyes with OCTA and under which circumstances the examination of flow within polypoidal lesions would be feasible.

The authors defined high flow lesions as those that with the presence of polyps on both ICGA and OCTA as determined by the examiners, on the other hand, low flow polyps were only detected on ICGA. Where the examiners for ICGA and OCTA blinded from each other? How many image graders were there? In case the examiners of ICGA and OCTA were not independent, would there be any bias ?

I suggest the authors to enhance the research methodology of the manuscript, and to provide more details on the intrinsic limitations of the qualitative study design. Furthermore, it would be interesting to know what would be the author’s point of view on the clinical implication of polyp regression. Both polyp regression and polyp inactivity had been reported in large, prospective clinical trials of PCV. Recurrence or persistence of polyp activity (e.g. exudation on scans) may influence on treatment decision. It seems that the definition of high/low flow signal OCTA in this manuscript is associated with partial or complete polyp ‘remission’. Please clarity if remission’ is regarding to the regression of polyp (i.e. change in size) and/or polyp activity (i.e. leakage)? Although polyp regression rate has been a reported outcome in a number of clinical trials, it may not be equivalent to disease activity. It would be intriguing to discuss the current evidence based practice with regard to polyp regression/activity, and to review the prognostic implication of polyp regression. To my knowledge, PLANET and EVEREST II had only published 2-year results. Please review any updated information on longer term PCV outcomes in the context of polyp remission.

Reviewer #3: The article is an excellent effort at bringing attention to the potential role of OCT Angiography flow signals in PCV polyps as a non-invasive predictive biomarker of its response to therapy. The authors have done a commendable job in endeavouring to correlate OCTA flow signals in PCV polyps to their ICGA behaviour, structural OCT features like PED height and subretinal fluid. While there are several limitations, that the authors have acknowledged, the study has numerous strengths as mentioned above and therefore merits a place in literature after the below stated minor errors are corrected.

Line 106 - µm should be used instead of um

Line 211 – PLANET study needs to be in caps.

Line 236 – spelling of polyps

Figure 2 – Lots of the boxes are blank.. needs to be clearly formatted

Case 2 in Table 1- mentions the location of the OCTA flow signal as Outer retina? That does not qualify for a PCV polyp OCTA flow signal and raises doubt as to whether it was a projected artefact. This is a critical error that needs to be corrected. Infact, the location of all OCTA PCV polyps needs to be beneath RPE ( All the other polyps here ,except this, have that). Ideally, the depth of the signal below the RPE could be specified. If that is not possible this aspect of the table may be omitted altogether as it adds no useful information.

6. PLOS authors have the option to publish the peer review history of their article (what does this mean?). If published, this will include your full peer review and any attached files.

Reviewer #1: Yes: Dr. Atul Kumar

Reviewer #2: No

Reviewer #3: Yes: Anand Rajendran

---

## [Author Response · Author response to Decision Letter 0]

23 Jun 2020

Dear Reviewer 1: 

Comment 1: Kindly explain why horizontal OCTA scans have been preferred over En-face OCTA scan

Response: The En-face OCTA and horizontal OCTA scans were both used for flow signal detection. The signal of En- face OCTA was studied for flow signal at areas of polyps and BVN shown in ICGA. To confirm the flow signal, especially in equivocal signal after anti-VEGF, horizontal OCTA scan were used across the polyps line by line for confirmation. This is why we set the definition of positive flow signal in 3 consecutive horizontal scans on the polyps. 

(This was further clarified in Page 5 line 108)

Comment 2: Presence of Branching vascular network (BVN) complex has not been recorded. Any correlation between polyps and BVN complex could also have been identified

Response: Thank you for the constructive advice. We had evaluated the presence of BVN in OCTA and ICGA, and found that 9 among 13 patients had BVN before the treatment. The size of BVN ranged from 0.75 to 3.0 disc area(DA) (average of 1.42±0.75 DA). After treatment, the size of the BVN remained unchanged in this short term 3 months follow up. This was in accordance with previous studies that the BVN had poorer response to anti-VEGF than the polyps did.

(This was further clarified in Table 1, Page 8 line 159, Page 9 line 167, Page 11 line 225)

Comment 3: Fig 1 (F, G, H) showing En-face OCTA of low flow polyp, could not be considered a proper scan. The manual segmentation is not at the appropriate level to be able to visualise the polyp.

Response: Since the case had prominent SRF, it was hard to visualize the polyp on the En-face OCTA even with manual segmentation. Fig 1 (F, G, H) explained the definition of low flow polyps in our study, of which no flow signal in 3 consecutive horizontal OCTA scan in the corresponding polyp location of the ICGA. There was an adjacent high flow polyp pointed out by arrow for comparison. 

(New figure 1 (revised F, G, H) had been submitted, and figure legends had been revised in Page 6 line 129.)

Comment 4: Figure 2 is incomplete. No details of high flow polyps given.

Response: Figure 2 is the flow chart of all the outcomes of polyps. The blanks might be due to format converting error and was probably incompletely uploaded. We will re-submit and upload again and will check with the editor office to ensure the completeness of the figure. Figure 2 had been re-submitted.

Comment 5: Figure 3 depicting ICGA after treatment, shows complete remission of polyp 1 and 3. However, flow signals in the polyps still persist on OCTA after treatment. How can it be explained?

Response: This is a very interesting point. We did see 2 among 8 polyps had completely regressed in ICGA without fluorescence after treatment yet still had flow signal in OCTA. Theoretically, the OCTA scan could detect very slow flow signal with velocity of 0.2 to 3 mm/s. Therefore, though the ICGA after the treatment revealed complete regression of the polyp, there might be slow flow remaining in the area that could be defected by the OCTA but not visualized in ICGA. 

This was further clarified in Page 13 line 258

Comment 6: A sample of 2 low flow polyps is clearly inadequate to conclude the relevance or to predict clinical response to anti-VEGF

Response: Thank you and we agreed with you that this is the limitation of our study with small numbers of cases. However, on the other hand, because of the wide detection of flow signal by OCTA (theoretically 0.2 to 3mm/second), the low flow polyps may indeed the polyps with very low filling flow of the polyps and the number may be quite few in symptomatic patients with PCV. It will be interesting to see if there are more low flow polyps in asymptomatic patients with PCV.

This was further clarified in Page 13 line 265

Dear Reviewer 2:

Comment 1: PCV eyes with poor visual acuity may not be able to fixate well during OCTA scan leading to motion artefacts. Were there any eyes excluded from the study and what were the reason for their exclusion? I believe providing details on the exclusion criteria and the number of patients that were excluded from the consecutive recruitment process is very important as it conveys to the readers how practical it would be to examine PCV eyes with OCTA and under which circumstances the examination of flow within polypoidal lesions would be feasible.

Response: The visual acuity of our naïve patients in this study ranged from 6/30~6/7.5. Patients with poor image quality due to poor fixation even with good visual acuity were exclude from the study. During this study period, there were 49 patients who met the exclusion criteria and were not enrolled in this study. These exclusion criteria included history of previous treatment with IVI or PDT, poor image quality due to fixation loss, medium opacity, massive hemorrhagic PED or submacular hemorrhage obscuring the polyps. 

(The exclusion criteria had been revised in line 81 and line 134)

Comment 2: The authors defined high flow lesions as those that with the presence of polyps on both ICGA and OCTA as determined by the examiners, on the other hand, low flow polyps were only detected on ICGA. Were the examiners for ICGA and OCTA blinded from each other? How many image graders were there?

Response: There were 2 retinal specialists to interpret each color, OCT and ICGA images separately. They were blind to each other’s results. Lesions were enrolled only when they were considered to be polyps by both retinal specialists. The en face OCTA and manual segmentation of each polyp was then studied by a third specialist. This third specialist interpret the signal of OCTA at the polyps where the 2 specialists agreed on.

This was further clarified in line 85, and line 105

Comment 3: Please clarity if remission’ is regarding to the regression of polyp (i.e. change in size) and/or polyp activity (i.e. leakage)?

Response: The term “remission” in the study referred to regression of the polyp in ICGA whether it was completely disappeared or decreased in size. This had been defined at the Materials & Methods, line 90. The decreased activity with decreased subretinal fluid or no fluid was not regarded as remission. The activity change, e.g., SRF, PED, had been described in Table 1. However, considering that regression was more commonly used and to avoid misunderstanding, we had revised by replacing “remission” with “regression”. Thank you.

(The “remission” and been revised to “regression” in the text in line 29, 35, 36, 90, 91, 170, 176, 183, 190, 199, 201, 205, 219, 221, 229, 234, 247, 253.)

Comment 4: Please review any updated information on longer term PCV outcomes in the context of polyp remission.

Response: A recent study with up to 11 years of follow-up showed that the long-term risk for massive submacular hemorrhage was significantly higher in patients with persistent polyps than patients without. (Chou et al, Retina 2020; 40:468-476.)

This long term FU study of PCV had been added and discussed in line 229.

Dear Reviewer 3:

Comment 1: Line 106 - µm should be used instead of um

Response: Thank you for pointing out our error. The correction had been made in page 6, line 114.

Comment 2: Line 211 – PLANET study needs to be in caps.

Response: Thank you for pointing out our error. The correction had been made at page 12, line 234.

Comment 3: Line 236 – spelling of polyps

Response: Thank you for pointing out our error. The correction had been made at page 13, line 264.

Comment 4: Figure 2 – Lots of the boxes are blank needs to be clearly formatted.

Response: Figure 2 is the flow chart of all the outcomes of polyps. The blanks might be due to format converting error and was probably incompletely uploaded. We will re-submit and upload again and will check with the editor office to ensure the completeness of the figure. Figure 2 would be re-submitted.

Comment 5: Case 2 in Table 1- mentions the location of the OCTA flow signal as Outer retina? That does not qualify for a PCV polyp OCTA flow signal and raises doubt as to whether it was a projected artefact. This is a critical error that needs to be corrected. In fact, the location of all OCTA PCV polyps needs to be beneath RPE (All the other polyps here ,except this, have that). Ideally, the depth of the signal below the RPE could be specified. If that is not possible this aspect of the table may be omitted altogether as it adds no useful information.

Response: We apologize for such error. The OCTA of the case was reviewed, and confirmed that the polyp was still beneath the RPE. Thank you for pointing out our mistake. 

(Table 1 and page 8 line 158 had been revised.)

---

## [Decision Letter · Decision Letter 1]

26 Aug 2020

PONE-D-20-09993R1

Flow Signal Change in Polyps After Anti-Vascular Endothelial Growth Factor Therapy

PLOS ONE

Dear Dr. Chen,

Thank you for submitting your manuscript to PLOS ONE. After careful consideration, we feel that it has merit but does not fully meet PLOS ONE’s publication criteria as it currently stands. Therefore, we invite you to submit a revised version of the manuscript that addresses the points raised during the review process.

ACADEMIC EDITOR: One of the reviewers have raised some persisting concerns; please do address them

We look forward to receiving your revised manuscript.

Kind regards,

Pukhraj Rishi

Academic Editor

PLOS ONE

Additional Editor Comments (if provided):

One of the reviewers have raised some persisting concerns; please address them

Reviewers' comments:

Reviewer's Responses to Questions

**Comments to the Author**

1. If the authors have adequately addressed your comments raised in a previous round of review and you feel that this manuscript is now acceptable for publication, you may indicate that here to bypass the “Comments to the Author” section, enter your conflict of interest statement in the “Confidential to Editor” section, and submit your "Accept" recommendation.

Reviewer #1: (No Response)

Reviewer #2: All comments have been addressed

2. Is the manuscript technically sound, and do the data support the conclusions?

Reviewer #1: Yes

Reviewer #2: Yes

3. Has the statistical analysis been performed appropriately and rigorously? 

Reviewer #1: Yes

Reviewer #2: Yes

4. Have the authors made all data underlying the findings in their manuscript fully available?

Reviewer #1: Yes

Reviewer #2: Yes

5. Is the manuscript presented in an intelligible fashion and written in standard English?

Reviewer #1: Yes

Reviewer #2: Yes

6. Review Comments to the Author

Reviewer #1: Figure 1 (F, G, H) images are not showing the segmentation line. The en-face OCTA scans do not show any part of vasculature. Even the adjacent high flow polyp is not showing flow signal on en-face OCTA. Can the authors provide any other representative ICGA and OCTA images of low flow polyp?

Reviewer #2: I would accept the revised version. The revised version had adequately answered my previous comments.

7. PLOS authors have the option to publish the peer review history of their article (what does this mean?). If published, this will include your full peer review and any attached files.

Reviewer #1: **Yes: **Atul Kumar

Reviewer #2: No

---

## [Author Response · Author response to Decision Letter 1]

1 Sep 2020

We thank the reviewer’s comments and suggestion. We had revised and provide the en-face image in figure 1 (F,G,H). The revised segmentation line with widened slab allow us to visualize the high flow and low flow polyps on en-face OCTA even they are at different height beneath the RPE with prominent subretinal fluid above. The 3 consecutive horizontal scan of the low flow polyp showed no flow signal comparing to the adjacent high flow polyp. 

The figure legend of figure 1 were also revised accordingly.

---

## [Editor Report · Decision Letter 2]

12 Oct 2020

Flow Signal Change in Polyps After Anti-Vascular Endothelial Growth Factor Therapy

PONE-D-20-09993R2

Dear Dr. Chen,

We’re pleased to inform you that your manuscript has been judged scientifically suitable for publication and will be formally accepted for publication once it meets all outstanding technical requirements.

Kind regards,

Alfred S Lewin, Ph.D.

Section Editor

PLOS ONE
---

## [Editor Report · Acceptance letter]

14 Oct 2020

PONE-D-20-09993R2 

Flow signal change in polyps after anti-vascular endothelial growth factor therapy 

Dear Dr. Chen:

I'm pleased to inform you that your manuscript has been deemed suitable for publication in PLOS ONE. Congratulations! Your manuscript is now with our production department. 

Kind regards, 

on behalf of

Dr. Alfred S Lewin 

Section Editor

PLOS ONE